**Data Availability Statement:** To protect the identification of the participants, some restrictions do apply to the primary data. These data can be

# Seroprevalence of SARS-CoV-2 infection and associated factors among Bangladeshi slum and non-slum dwellers in pre-COVID-19 vaccination era: October 2020 to February 2021

**Rubhana Raqib**[1]*, **Protim Sarker**[1], **Evana Akhtar**[1], **Tarique Mohammad Nurul Huda**[1], **Md. Ahsanul Haq**[1], **Anjan Kumar Roy**[1], **Md. Biplob Hosen**[1], **Farjana Haque**[1], **Md. Razib Chowdhury**[2], **Daniel D. Reidpath**[2], **Dewan Md. Emdadul Hoque**[3], **Zahirul Islam**[4], **Shehlina Ahmed**[5], **Tahmeed Ahmed**[6], **Fahmida Tofail**[6], **Abdur Razzaque**[2]

1 Infectious Diseases Division, icddrb, Dhaka, Bangladesh, 2 Health Systems and Population Studies Division, icddrb, Dhaka, Bangladesh, 3 United Nations Population Fund (UNFPA), Dhaka, Bangladesh, 4 Embassy of Sweden, Dhaka, Bangladesh, 5 Foreign, Commonwealth & Development Office (FCDO), Dhaka, Bangladesh, 6 Nutrition and Clinical Services Division, icddrb, Dhaka, Bangladesh

* rubhana@icddrb.org

## Abstract

### Background

Seroprevalence studies have been carried out in many developed and developing countries to evaluate ongoing and past infection with severe acute respiratory syndrome coronavirus 2 (SARS-CoV-2). Data on this infection in marginalized populations in urban slums are limited, which may offer crucial information to update prevention and mitigation policies and strategies. We aimed to determine the seroprevalence of SARS-CoV-2 infection and factors associated with seropositivity in slum and non-slum communities in two large cities in Bangladesh.

### Methods

A cross-sectional study was carried out among the target population in Dhaka and Chattogram cities between October 2020 and February 2021. Questionnaire-based data, anthropometric and blood pressure measurements and blood were obtained. SARS-CoV-2 serology was assessed by Roche Elecsys® Anti-SARS-CoV-2 immunoassay.

### Results

Among the 3220 participants (2444 adults, ≥18 years; 776 children, 10–17 years), the overall weighted seroprevalence was 67.3% (95% confidence intervals (CI) = 65.2, 69.3) with 71.0% in slum (95% CI = 68.7, 72.2) and 62.2% in non-slum (95% CI = 58.5, 65.8). The weighted seroprevalence was 72.9% in Dhaka and 54.2% in Chattogram. Seroprevalence was positively associated with limited years of formal education (adjusted odds ratio [aOR] = 1.61; 95% CI = 1.43, 1.82), lower income (aOR = 1.23; 95% CI = 1.03, 1.46), overweight

made available from the Ethics Committees (ERC/RRC) at the International Centre for Diarrhoeal Disease Research, Bangladesh (icddr,b) for researchers who meet the criteria for access to confidential data; please contact the Head of Research Administration at the icddr,b (Armana Ahmed; aahmed@icddrb.org).

**Funding:** This work was funded by The Foreign, Commonwealth & Development Office (FCDO) through The United Nations Population Fund (UNFPA), and Global Affairs Canada. icddr,b is also grateful to the Governments of Bangladesh, Canada, Sweden and the UK for providing core/unrestricted support for its operations and research.

**Competing interests:** The authors have declared that no competing interests exist.

(aOR = 1.2835; 95% CI = 1.26, 1.97), diabetes (aOR = 1.67; 95% CI = 1.21, 2.32) and heart disease (aOR = 1.38; 95% CI = 1.03, 1.86). Contrarily, negative associations were found between seropositivity and regular wearing of masks and washing hands, and prior BCG vaccination. About 63% of the population had asymptomatic infection; only 33% slum and 49% non-slum population showed symptomatic infection.

## Conclusion

The estimated seroprevalence of SARS-CoV-2 was more prominent in impoverished informal settlements than in the adjacent middle-income non-slum areas. Additional factors associated with seropositivity included limited education, low income, overweight and pre-existing chronic conditions. Behavioral factors such as regular wearing of masks and washing hands were associated with lower probability of seropositivity.

## Introduction

The coronavirus disease 2019 (COVID-19), caused by severe acute respiratory syndrome coronavirus 2 (SARS-CoV-2) led to immense suffering and an unprecedented burden on health care systems globally. It was declared a 'pandemic' by the World Health Organization (WHO) in March 2020. Worldwide, numbers of confirmed COVID-19 cases are based on PCR-positivity of nasopharyngeal samples for SARS-CoV-2 among symptomatic subjects. However, reported cases and deaths especially in low and middle income countries (LMIC), are likely to be underestimates of the actual disease prevalence; diverse clinical presentations like asymptomatic infections, limited testing capacity, hesitation and inaccessibility to test are key contributing factors. The WHO has recommended all countries to carry out population-based SARS-CoV-2 seroprevalence surveys for rapid screening of ongoing epidemics, determining the extent of spread and estimating the proportion of symptomatic and asymptomatic infected subpopulations [1–3]. Population-based serosurveys for SARS-CoV-2 conducted in LMICs have been either national surveys [4,5] or predominantly focused on frontline healthcare workers or industry workers [6–10] but few have studied marginalized populations in slums [11,12].

People living in urban slums are more susceptible to multiple health threats than non-slum populations because of their underprivileged living conditions. Consequently, the population becomes a major reservoir for a wide spectrum of diseases, which may accelerate the spread of infection [13,14]. Commonly perceived risk factors include population density, poverty, lack of educaton, lack of access to clean water and soap, poor hygiene and hand-washing practices, inability to maintain physical distance, and limited access to health care. Slums in the developing world e.g. Mumbai, Rio de Janeiro, Lagos as well as marginalized city population in resource-rich countries e.g. New York and London have witnessed rapid spread of COVID-19 infections with disastrous consequences [15–19]. Around 12.2 million impoverished people of Dhaka and Chattogram live in densely populated, low-income urban slums [20–22]. So far limited data about the COVID situation in these slum communities are available from Bangladesh [23]. Studies targeting RT-PCR-based detection of SARS-CoV-2 infection in Bangladesh have been mainly carried out in hospital settings [24–26]; such data from the community is lacking. COVID-19 seroprevalence surveys are helpful to estimate the percentage of people in the communities who have been previously infected with SARS-CoV-2 and may reflect the extent of spread of the infection.

Older age, male sex, overweight and underlying chronic diseases have been found to be associated with severity of COVID-19 [27,28]. It is important to find out whether different demographic characteristics and co-morbidities prevailing in the slum and non-slum urban areas are also associated with seroprevalence of SARS-CoV-2. The findings can help in determining targeted public health measures to address the specific requirements of the populations most in need.

We aimed to study the seroprevalence of SARS-CoV-2 to evaluate the magnitude of spread of infection in the poor slum and surrounding non-slum middle-class communities in Dhaka, and Chattogram, the two large cities of Bangladesh, and identify factors associated with seropositivity.

## Methods

### Study design and setting

The survey was conducted in purposively selected large slums and neighboring non-slum areas of Dhaka and Chattogram from October 2020 to February 2021. A slum is defined here as a cluster of compact settlements of 5 or more households, which generally grow unsystematically and haphazardly in an unhealthy condition and atmosphere, on government and private vacant land [29]. In 2015, the icddr,b established an urban Health and Demographic Surveillance System (UHDSS) in Dhaka over 125,000 population to include slums. The location, household head's name, age, identification number, and GIS coordinates were available from the Dhaka UHDSS [30]. Four slums in and around Dhaka city (Korail, Mirpur, Dhalpur and Ershad Nagar) under UHDSS were selected. Slum Census Report 2014, Bangladesh Bureau of Statistics [29] was used for selecting 2 slums in Chattogram city (Shaheed Lane and Akbar Shah Kata Pahar). The non-slum area was selected to include middle-class households, defined as single or multi-stored buildings with wall and roof constructed with bricks, and having one or no security guard in the building.

For selecting the sample, cluster random sampling procedure was followed. After mapping the study area, households were divided into clusters (equal size) and clusters were selected randomly to get the required sample size (Dhaka: 15 clusters from slum, 12 clusters from non-slum; Chattogram: 12 clusters for slum, 8 clusters for non-slum). UHDSS sampling frame was used for selecting the household in Dhaka slums. For rest of the areas, household listing was done initially for the selected clusters, and data was collected according to the list. Please see S1 File for field data collection and study team.

### Sampling frame

Data and samples were collected from 2118 slum inhabitants and 1102 from non-slums in a 2:1 ratio. After household selection, all the eligible members aged $\geq$10 years, were invited to participate. In the early phase of the pandemic, susceptibility to COVID-19 was rare in <10 years old children and existing reports on adolescents (10–19 years) were controversial [31]. Therefore, adolescents, who can give assent and donate blood were enrolled in this study to determine seroprevalence of SARS-CoV-2 in this age group along with the adults. The exclusion criteria included refusal to give time to respond to all questions and give blood. For sample size calculation please see S2 File.

### Household questionnaire survey

Participants were enrolled in the study after obtaining informed consent/assent. The household questionnaire included data on (i) sociodemographic features, (ii) preventive behavior

practiced within last 6 months, (iii) chronic health conditions, (iv) current morbidity or in the past 6 months, related to COVID-like symptoms, (v) presence of confirmed COVID-19 cases in the household and (vi) physical activity assessment using Global Physical Activity Questionnaire (GPAQ) version 2 [32]. The GPAQ consists of 3 parts: (a) work, (b) travel to and from places, and (c) recreational activities. However, only sections (a) and (b) were included in this study to reduce the total time of data collection. The physical activity was categorized into vigorous-, moderate- and light- intensity activities (walking or cycling to visit places in a typical day). For physical activity assessment, 2012 individuals were included as the GPAQ assessment tool was introduced few weeks later in the study. BCG vaccination status was determined by checking the immunization card or verifying the vaccination mark on the upper arm.

The sociodemographic data included age, sex, education, migration background, marital status, employment, and income. The family's monthly income was considered as a proxy indicator of its economic status. The preventive behavior included wearing masks, washing hands with soap, wearing gloves, sneezing on hand or tissue paper, maintaining a distance of 6 feet while meeting others, avoiding crowds, avoiding putting fingers in the nose or on the face, repeated drinking of hot water. Chronic health conditions assessed included diabetes, asthma, lung disease, hypertension, heart disease, stroke and cancer. These were confirmed by checking physician's prescription or medical report. The self-reported COVID-like symptoms included fever or chills, cough, sore throat, congestion or runny nose, shortness of breath or difficulty in breathing, fatigue, muscle or body aches, headache, loss of taste or smell, nausea or vomiting and diarrhea [33]. Confirmed COVID-19 cases included those who were identified as positive by PCR for SARS-CoV-2 in nasopharyngeal samples (by checking test report).

The survey data collected in Tablet/Android-based electronic questionnaire was synchronized with the server after each interview. The SQLite program was used to write data collection, SQLite browser, SQL and Visual Foxpro were used for data management and data cleaning.

## Anthropometry, blood pressure, specimen collection

Height and weight were measured twice using the free-standing stadiometer (Seca 217, Hamburg, Germany) and digital weighing scale (Camry-EB9063, China) calibrated weekly, and body mass index (BMI) was calculated. Blood pressure (BP) was measured twice using a manual BP machine (ALPK2 V500, Japan) in a sitting position with a 5-minutes interval between the measurements, and the average was used. The study defined hypertension was, systolic blood pressure $\geq$140 mmHg, or diastolic blood pressure $\geq$90 mmHg. Single venous blood samples (7.5 ml) were collected in trace element-free heparinized tubes in the household and were delivered to the Laboratory within 2–3 hours. Plasma was separated and stored at -80˚C until analysis.

## Assessment of SARS-CoV-2 specific antibodies

The Elecsys® Anti-SARS-CoV-2 assay was used to determine the nucleocapsid (N) antigen-specific antibodies (IgM and IgG) against SARS-CoV-2 in plasma on Cobas-e601 immunoassay analyzer (Roche Diagnostics GmbH, Mannheim) indicating recent or prior infection. Based on the antibody cut-off index (COI), the serological response to SARS-CoV-2 is categorized as reactive (COI$\geq$1.0, seropositive) and non-reactive (COI<1.0, seronegative). According to the manufacturer's performance characteristics, the Elecsys assay has an overall specificity of 99.8% (95% CI = 99.69–99.88%) and an overall sensitivity of >99.5% (95% CI = 97.0–100%) for $\geq$14 days of post PCR confirmation. We carried out an internal validation to evaluate the kit's performance. In this study, the assay showed an overall specificity of 100%

(95% CI, 97.9%-100%) and an overall sensitivity of >93.3% using serum samples from SARS-CoV-2 specific RT-PCR positive cases (n = 30x3 from day 7, 14 and 21 days post diagnosis and n = 70 from day >21; total 160), pre-pandemic COVID-19 negative healthy cases (n = 100), non-bacterial pneumonia cases (n = 51) and other viruses (n = 13) (S1 Table).

### Ethics approval

The study was approved by the institutional review board (PR-20070, dated 1[st] September 2020) of the icddr,b. Written informed consent was obtained from adult participants while assent was obtained from 10–17 years old children and consent from their parents.

### Statistical analysis

The main outcome of interest SARS-CoV-2 serological data was categorized as reactive (seropositive) and non-reactive (seronegative) based on the antibody cut-off. Demographic data was expressed as number and percentages for categorical observation or mean with standard deviation for continuous data. The prevalence of seropositivity among the demographic features, body mass index (BMI), COVID-like symptoms, co-morbidities, preventive measures practiced and physical activities, stratified by locality (slums, non-slums) was expressed as prevalence with 95% CI. A population-based weighted seroprevalence of SARS-CoV-2 was calculated on the basis of sum of two probabilities i.e. between cluster probability (p1) and within cluster probability (p2). The sample was collected from 4 areas of Dhaka (Korail, Mirpur, Dhalpur and Ershad Nagar) and 2 areas of Chottagram (Shaheed Lane and Akbar Shah Kata Pahar) that included both slum and neighboring non-slum areas. For each selected area, a total probability score was calculated (p1+p2). The weight was determined as the inverse of total probability of each selected area (1/(p1+p2)). Thereafter, the calculated weight was distributed to all selected participants and the weighted prevalence was estimated.

Initially, univariate logistic regression was performed to estimate the relationship between several predictors (sociodemographic factors, BMI, Bacillus Calmette-Guérin (BCG) vaccination, symptoms, comorbidities, preventive measures practiced and physical activities) and seropositivity. Since the in-house validation of the antibody assay showed a sensitivity of about 93% and a specificity between 97.9%-100% (manufacturer reported sensitivity is 99% and specificity is 100%), to correct the test inaccuracy, we estimated the seroprevalence of SARS-CoV-2 associated risks (odds ratio) by Bayesian multivariate generalized linear mixed model (MGLMM). The Bayesian MGLMM was fitted with sociodemographic factors, BMI, Bacillus Calmette-Guérin (BCG) vaccination, symptoms, comorbidities, preventive measures practiced and physical activities as fixed effects and cluster effects (weighted score) were taken as a random effect. The data management and statistical analyses were performed with Stata 15 (StataCorp, LP, College Station, Texas, USA), and the graphs were prepared by GrapPad prism 8.3.0. The significance level was established at $p \leq 0.05$.

## Results

### Demography

Data were collected from 3,220 inhabitants of 1337 households (1910 from Dhaka slum, 705 from Dhaka non-slum, 334 from Chattogram slum and 272 from Chattogram non-slum) with an average of 4 members per household. In Dhaka, around 64% of slum and 34% of non-slum populations agreed to provide data and blood samples, while in Chattogram the respondents were 69% for slum and 44% for non-slum. The majority of the middle-class families from non-slum areas did not allow study staffs to enter their households for fear of getting infected

**Table 1. Demographic characteristic of the study participants.**

| Variables | | Overall (n = 3220) | Slum (n = 2118) | Non-slum (n = 1102) |
|---|---|---|---|---|
| Sex | | | | |
| | Male | 1392(43.2%) | 953(44.8%) | 439(40.2%) |
| | Female | 1828(56.8) | 1175(55.2%) | 653(59.8%) |
| Age, years | | | 31.02±16.42 | 33.34±16.63 |
| Age, category | | | | |
| | 10–17 years | 776(24.1%) | 555(26.1%) | 221(20.2%) |
| | 18–30 years | 945(29.4%) | 623(29.3%) | 322(29.4%) |
| | 31–50 years | 1015(31.5%) | 645(30.4%) | 370(33.8%) |
| | >50 years | 484(15.0%) | 302(14.2%) | 182(16.6%) |
| House hold member, mean±SD | | 4.87±1.95 | 4.76±1.92 | 5.09±2.0 |
| Education in years | | | | |
| | No education | 847(26.3%) | 754(35.4%) | 93(8.52%) |
| | 1–5 years | 966(30.0%) | 804(37.8%) | 162(14.8%) |
| | 6–10 years | 942(29.3%) | 500(23.6%) | 442(40.3%) |
| | 11–15 years | 465(14.4%) | 70(3.29%) | 395(36.2%) |
| Occupation | | | | |
| | Service | 469(14.6%) | 310(14.6%) | 159(14.6%) |
| | Self employed | 336(10.4%) | 294(13.9%) | 42(3.83%) |
| | Business | 262(8.14%) | 152(7.16%) | 110(10.0%) |
| | Homemakers | 846(26.3%) | 488(22.9%) | 358(32.8%) |
| | Unemployed | 504(15.7%) | 399(18.8%) | 105(9.62%) |
| | Student | 803(24.9%) | 485(22.8%) | 318(29.1%) |
| Monthly income, taka | | | | |
| | <20000 | 1280(39.8%) | 1241(58.5%) | 39(3.56%) |
| | 20000–40000 | 1019(31.7%) | 767(36.15) | 252(23.0%) |
| | 40000–70000 | 636(19.8%) | 115(5.42%) | 521(47.5%) |
| | >70000 | 285(8.85%) | 5(0.23%) | 280(25.6%) |
| Presence of COVID-19 like symptoms | | 1144(35.5%) | 673(31.7%) | 471(42.9%) |
| BCG given | | 2717(84.4%) | 1766(83.2%) | 951(86.7%) |
| BMI | | 23.43±5.24 | 22.65±5.07 | 24.94±5.23 |
| | Normal | 1405(43.6%) | 971(45.6%) | 434(39.7%) |
| | Underweight | 607(18.9%) | 489(23.0%) | 118(10.8%) |
| | Overweight | 1208(37.5%) | 668(31.4%) | 544(49.5%) |

Note. BMI, Body mass index; BCG, Bacillus Calmette-Guérin. Data was presented as mean±SD or number (percent).

with SARS-CoV-2. The demographic features of the population in slums and non-slums are given in Table 1. The male, female distribution among the enrolled participants was 43% and 57%, respectively. Higher numbers of females were enrolled because of their availability at home during the visits; male members were mostly away for earning wages. The average age in slums and non-slum areas were 31 and 33.3 years, respectively with 24% being pre-/adolescents and 15% being above 50 years of age in the total study population. An average of 4.87 members were living in the same household in slums and 5.09 in nonslum areas. The percentage of inhabitants with longer years of formal education (11–15 years), monthly income of >40000 BDT (equivalent to USD $472) and BCG vaccine coverage were higher in non-slums than in slums. (Table 1). Around two-third of the seropositive participants (~63%) were asymptomatic. The proportion of asymptomatic infection was higher in seropositive

adolescents (71%) than adults (60%). More non-slum (49.3%) than slum populations (33%) experienced COVID-19 like symptoms.

## Seroprevalence of SARS-CoV-2 and sociodemographic features

The overall weighted seroprevalence of SARS-CoV-2 in the population was 67.3%, with a higher positivity rate among slum dwellers (71.0%) than the non-slum dwellers (62.2%) (Table 2). A higher weighted seroprevalence was observed in Dhaka city (72.9%) than seen in Chattogram (54.2%). Age-wise weighted seroprevalence rate among slum and non-slum dwellers of Dhaka and Chattogram cities is given in S2 Table.

Female participants of the study showed higher odds of seropositivity than males. The odds of seropositivity was higher among inhabitants who had no or <10 years of education compared to those with >11 years of education (Table 3). Households members having monthly family income of more than 70,000 BDT (equivalent to USD >$825) had lower seroprevalence compared to those with lower income (Table 2). Overweight individuals revealed higher SARS-CoV-2 seroprevalence 72.7% (95% CI 69.5%-75.7%) compared to those with normal BMI (range 18.5–24.9) (Table 2) and the odds was 1.35 (95% CI 1.26–1.97) folds higher in overweight individuals (Table 3). The seroprevalence was found to be lower in the BCG vaccinated 66.3% (95% CI 64%-68.6%) than non-vaccinated 71.4% (95% CI 66.3%-76%) participants

## Seroprevalence of SARS-CoV-2 and self-reported COVID-like symptoms/chronic diseases

About 36% of the household members reported the presence of ongoing COVID-like symptoms or occurrence within past 6 months. Overall individuals reporting fever, dry cough or sore throat had higher prevalence of SARS-CoV-2 infection (S3 Table). Additionally higher odds of seropositivity were obtained in the overall population having fever, sore throat and diarrhoea compared to those without symptoms (Fig 1). The odds of seropositivity was 1.55 (95% CI 1.30,1.86) folds higher for residents who exhibited presence of any 3 symptoms in the preceding 6 months compared to those without symptoms (Fig 1A). Only 4 slum dwellers and 18 non-slum participants reported that they were hospitalized with moderate to severe disease in the past 6 months and were seropositive for SARS-CoV-2.

Among the individuals reporting various co-morbid conditions, those with diabetes and heart diseases had higher seroprevalence of SARS-CoV-2 (77.5% (95% CI 71.%-82.9%) and 74.1% (95% CI 63.9%- 82.2%) respectively) (S4 Table). The individuals having diabetes and heart problem also had higher odds of seropositivity compared to those without the pre-existing chronic diseases (Fig 2). Only in non-slum participants, the odds of seropositivity was higher among those who had a history of stroke (aOR = 1.84; 95% CI 1.19,2.83) (Fig 2C). SARS-CoV-2 seropositivity was higher in slums among those participants who had a history of hypertension than those who did not [78.9% (95% CI 72.8%-84.0%) vs 71.8% (95% CI 68.9%-74.6%)] but not in non-slums (S4 Table). No other co-morbidities were associated with seroprevalence of SARS-CoV-2 in this population.

## Seroprevalence of SARS-CoV-2 and behavioral aspects/physical activity

Individuals who wore face masks on a regular basis had lower odds of getting seropositive (aOR = 0.33; 95% CI = 0.22, 0.46), which was evident in both slum and non-slum populations (Fig 3). Individuals who practiced washing hands with soap had lower odds of becoming seropositive (aOR = 0.40; 95% CI = 0.23, 0.73); this was more prominent in non-slum population (aOR = 0.19; 95% CI = 0.10, 0.35). No other personal beheviour of the participants were

**Table 2. Weighted seroprevalence of SARS-CoV2 among the residents of slum and non-slum neighborhoods.**

| Variables | Overall (n = 3220) | Slum (95% CI) (n = 2123) | Non-slum (95% CI) (n = 1097) |
|---|---|---|---|
| Overall | 67.3(65.2, 69.3) | 71.0(68.7, 72.2) | 62.2(58.5, 65.8) |
| Sex | | | |
| Male | 64.6(61.4, 67.6) | 67.0(63.4, 70.5) | 60.7(54.8, 66.3) |
| Female | 69.3(66.7, 71.9) | 74.5(71.4, 77.1) | 63.2(58.4, 67.9) |
| Age category, years | | | |
| 10–17 years | 62.8(58.5, 66.9) | 65.9(61.1, 70.5) | 56.9(48.5, 65.0) |
| 18–30 years | 68.0(64.2, 71.6) | 72.4(68.2, 76.2) | 62.2(55.2, 68.7) |
| 31–50 years | 69.7(66.1, 73.1) | 74.1(70.1, 77.8) | 64.3(57.8, 70.4) |
| > 50 years | 68.4(62.9, 73.4) | 71.3(64.9, 76.9) | 65.5(56.4, 73.6) |
| Years of education | | | |
| No education | 68.0(64.0, 71.8) | 71.6(67.8, 75.1) | 52.6(39.9, 64.9) |
| 1–5 years | 70.3(66.6, 73.7) | 71.3(67.3, 74.9) | 66.8(57.1, 75.3) |
| 6–10 years | 66.2(62.3, 69.9) | 69.2(64.3, 73.6) | 63.8(57.8, 69.3) |
| 11–15 years | 63.0(57.3, 68.3) | 75.2(61.9, 84.9) | 61.3(55.1, 67.1) |
| Occupation | | | |
| Service | 72.5(67.3, 77.3) | 78.7(73.3, 83.3) | 64.4(54.5, 73.3) |
| Self employed | 65.4(59.0, 71.4) | 67.2(60.7, 73.1) | 56.7(36.9, 74.6) |
| Business | 66.3(58.9, 73.0) | 73.0(64.8, 80.0) | 59.4(47.1, 70.6) |
| Homemaker | 70.8(66.8, 74.5) | 77.1(72.6, 81.0) | 65.1(58.5, 71.1) |
| Unemployed | 65.0(59.7, 70.0) | 67.6(62.1, 72.7) | 59.6(47.4, 70.6) |
| Student | 63.1(58.9, 67.2) | 65.2(60.0, 70.2) | 60.6(53.7, 67.2) |
| Monthly income, taka | | | |
| <20000 | 67.5(64.4, 70.5) | 67.3(64.2, 70.3) | 70.8(53.4, 83.7) |
| 20000–40000 | 71.6(68.1, 74.8) | 76.3(72.7, 79.5) | 62.5(55.0, 69.5) |
| 40000–70000 | 64.3(59.3, 69.0) | 76.9(67.6, 84.1) | 62.6(57.1, 67.8) |
| >70000 | 58.4(50.7, 65.7) | - | 58.9(51.1, 66.3) |
| BMI | | | |
| Normal | 66.1(62.9, 69.1) | 71.7(68.3, 74.9) | 57.8(53.8, 63.6) |
| Underweight | 60.0(55.1, 64.7) | 60.6(55.3, 65.6) | 58.4(47.0, 68.9) |
| Overweight | 72.7(69.5, 75.7) | 79.0(75.4, 82.1) | 67.3(62.1, 72.0) |
| BMI (Adult) | | | |
| Normal | 66.3(62.7, 69.8) | 71.6(67.7, 75.1) | 58.4(51.4, 65.0) |
| Underweight | 62.0(53.8, 69.7) | 61.1(52.4, 69.1) | 64.3(45.6, 79.5) |
| Overweight | 72.6(69.3, 75.7) | 78.6(74.9, 81.9) | 67.4(62.1, 72.4) |
| BMI (Adolescent) | | | |
| Normal | 65.1(58.2, 71.5) | 72.6(64.5, 78.7) | 56.0(43.6, 67.6) |
| Underweight | 58.9(52.8, 64.7) | 60.4(53.8, 66.6) | 54.5(40.3, 68.0) |
| Overweight | 73.8(61.3, 83.4) | 82.8(67.0, 91.9) | 65.4(45.7, 80.9) |
| BCG vaccination | | | |
| Given | 66.3(64.1, 68.6) | 69.8(67.2, 72.2) | 62.0(57.9, 65.9) |
| Not given | 71.4(66.3, 76.0) | 76.1(70.8, 80.7) | 63.6(53.4, 72.7) |

Note. BMI, Body mass index; BCG, Bacillus Calmette-Guérin. Results was presented as prevalence with 95% confidence interval.

associated with the odds of seropositivity. Assessment of data on physical activities revealed that there was no impact of light-to-intense physical activity on SARS-CoV-2 seropositivity (data not shown).

**Table 3. Odds of seropositivity among the study participants residing in slum and non-slum areas.**

| | Overall | Slum | Non-slum |
|---|---|---|---|
| **Variables** | **OR(95% CI)** | **OR (95% CI)** | **OR(95% CI)** |
| Sex | | | |
| Male | Ref. | Ref. | Ref. |
| Female | 1.62(1.40, 1.86) | 1.70(1.01, 2. 94) | 1.25(0.87, 1. 73) |
| Years of education | | | |
| 11–15 years | Ref. | - | Ref. |
| 6–10 years | 1.47 (1.16, 1.88) | Ref. | 1.32(1.02, 1. 75) |
| 1–5 years | 1. 45(1.05, 11.99) | 120(0. 76, 1. 95) | 1.39(0.96, 1.70) |
| No education | 0.86 (0.62, 1.19) | 0. 87(0.44, 1.70) | 0.58 (0. 34, 1.03) |
| Monthly family income, taka | | | |
| >70000 | Ref. | - | Ref. |
| 40000–70000 | 1.35 (1.08, 1.72) | Ref | 1.39(1.09, 1.86) |
| 20000–40000 | 1. 28(0. 97, 1.63) | 1.47(0.70, 3.42) | 1.21(0.96, 1.70) |
| <20000 | 1.13(0.85, 1. 48) | 2.43(1.20, 5.21) | 1.35(0.86, 2.23) |
| BMI | | | |
| Normal | Ref. | Ref. | Ref. |
| Underweight | 0.97(0.75, 1.27) | 0.73(0.56, 0.94) | 1.28 (0.81, 2.05) |
| Overweight | 1.35(1.26, 1. 97) | 1.28(1.01, 1.63) | 1.39 (1.05, 1.79) |
| BCG vaccination | | | |
| Not given | Ref. | Ref. | Ref. |
| Given | 0.84(0.60, 0.96) | 0.79(0.48, 1. 17) | 0.80 (0.51, 1.35) |

Data was presented as Odds ratio (OR) with 95% confidence interval. Bayesian multivariate generalized linear mixed model was applied to estimate seroprevalence-associated risks (odds ratio). The regression model was adjusted by sex, age, years of education, occupation, family income and body mass index (BMI).

## Discussion

The study provides important insight into COVID-19 pandemic in the informal settlements and the urban neighborhood communities of two large cities in Bangladesh. This cross-sectional serosurvey involving more than 3,200 participants showed an overall weighted SARS-CoV-2 seroprevalence of 67.3%, with the seropositivity rate being higher in slums (71.0%) than in non-slum localities (62.2%). The significant factors associated with seroprevalence of SARS-CoV-2 among the study population included education, income, certain preventive behaviors, BCG vaccination status and pre-existing chronic conditions, such as diabetes, overweight, heart problems and stroke.

A large SARS-CoV-2 seroprevallence study carried out in India showed a seroprevalence of about 26% in 70 districts [8]. This study, included rural areas, in which one might anticipate lower prevalence because of the lower population density, and it was also conducted much earlier in the epidemic. Cross-sectional studies carried out in large Indian cities of states like Odisha, Madhya Pradesh, Karnataka and Maharashtra showed wide variation in seroprevalence between the cities, which increased with time in subsequent rounds (5% to 76.8%) [34–36]. The nation wide SARS-CoV-2 seroprevalence study in India carried out in August to September, 2020 showed a seroprevalence of 7%, while in June-July 2021, it increased to 67.6% [5,8]. The seroprevalence of SARS-CoV-2 we found towards the last stage of the first wave of COVID-19 in Bangladesh (October 2020-February 2021) before initiation of vaccination against COVID-19, was comparable to those seen in Karnataka and Maharashtra and in

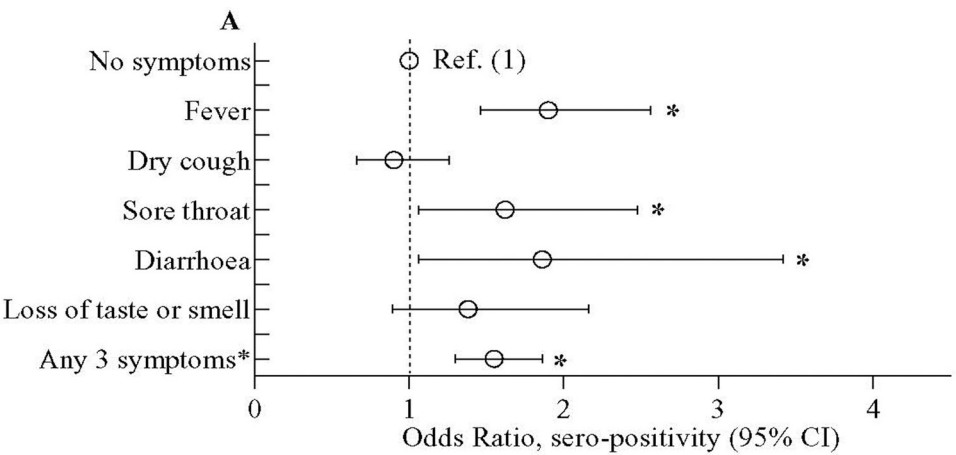

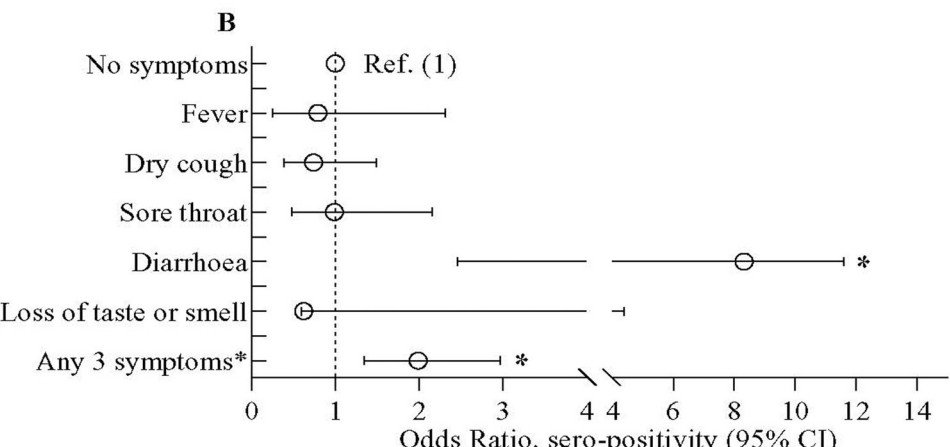

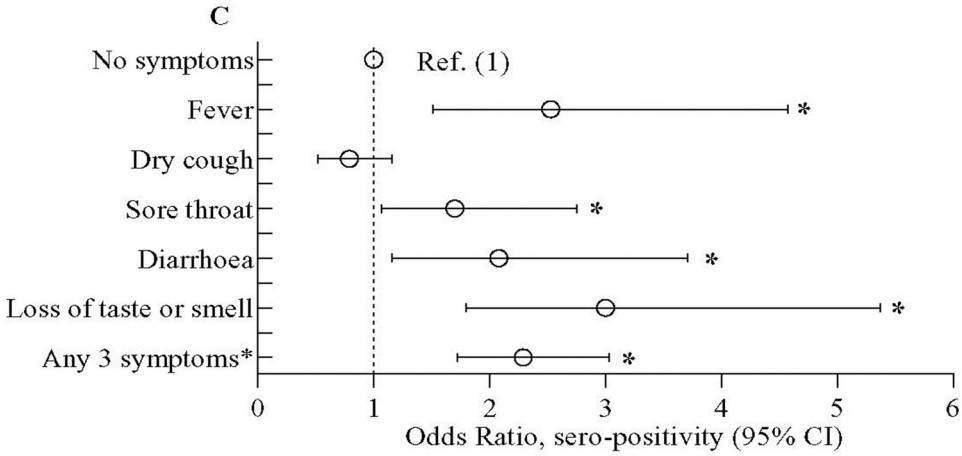

**Fig 1. Odds of SARS-CoV-2 sero-positivity among study participants with or without self-reported presence of COVID-19 like symptoms (*33*).** (A) represents data in overall study population, (B) on urban slum population and (C) on inhabitants of the neighboring non-slum areas. Data were presented as adjusted odds ratio (aOR) with 95% confidence interval. Bayesian multivariate generalized linear mixed model (MGLMM) was used to estimate the p-value. The Bayesian MGLMM was fitted with sociodemographic factors, BMI, Bacillus Calmette-Guérin (BCG) vaccination, and symptoms, as fixed effects and cluster effects were taken as a random effect.

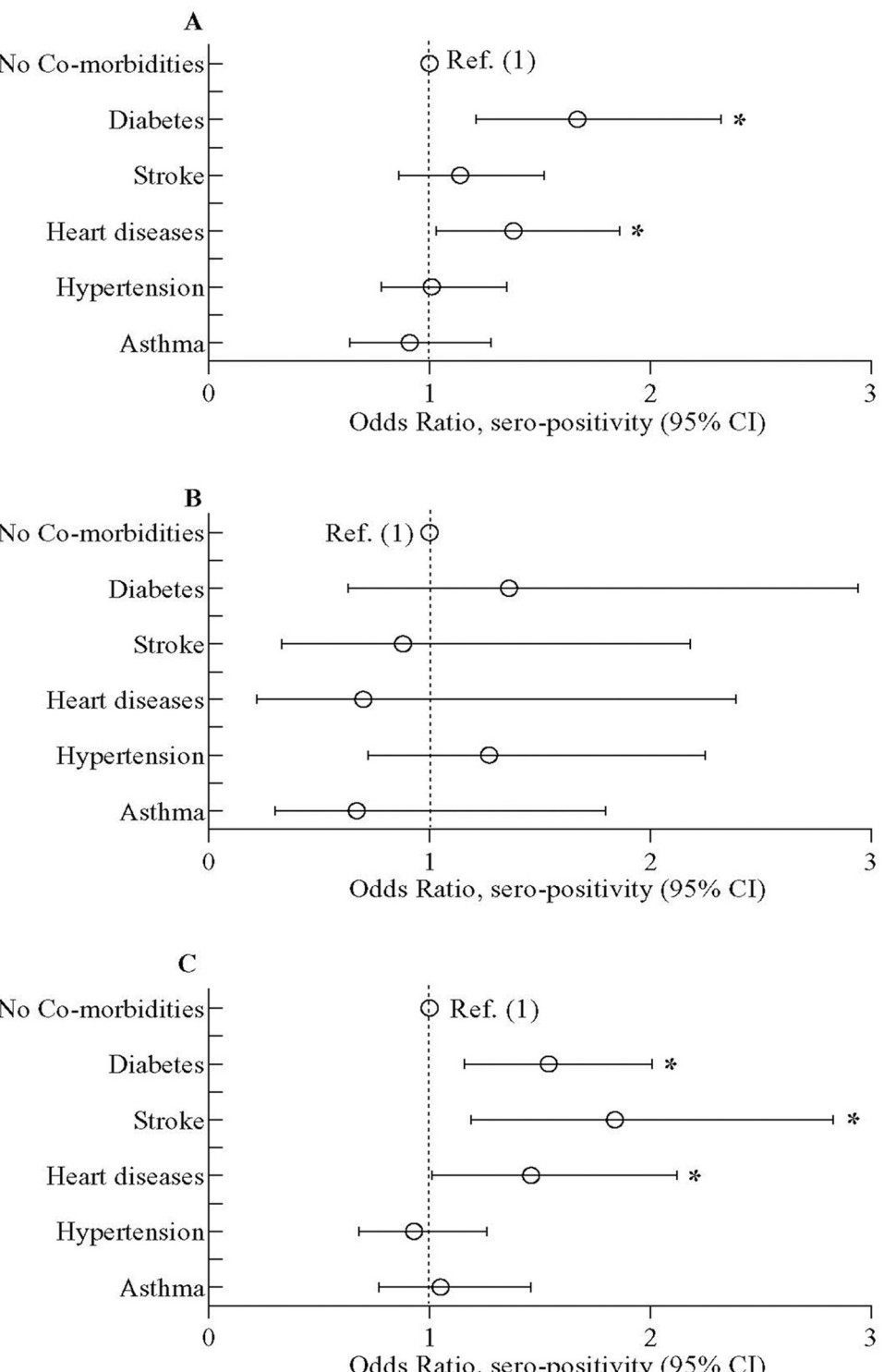

**Fig 2. Odds of SARS-CoV-2 sero-positivity among study participants with or without co-morbid conditions (diabetes, stroke, heart disease, hypertension, and asthma).** (A) represents data in overall study population, (B) on urban slum population and (C) on inhabitants of the neighboring non-slum areas. Data was presented as adjusted odds ratio (aOR) with 95% confidence interval. Bayesian multivariate generalized linear mixed model (MGLMM) was used to estimate the p-value. The Bayesian MGLMM was fitted with sociodemographic factors, BMI, Bacillus Calmette-Guérin (BCG) vaccination, and comorbidities, as fixed effects and cluster effects were taken as a random effect.

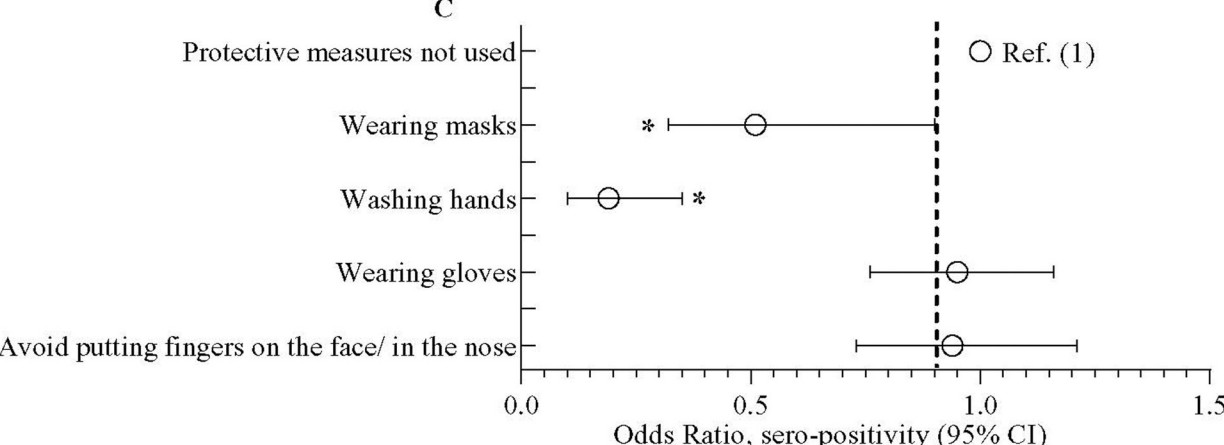

**Fig 3. Odds of SARS-CoV-2 sero-positivity among study participants who have taken or not taken preventive measures.** The preventive procedures include wearing masks, washing hands with soap, wearing gloves, avoid putting fingers in the nose and/ on the face. (A) represents data in overall study population, (B) on urban slum population and (C) on inhabitants of the neighboring non-slum areas. Data was presented as adjusted odds ratio (aOR) with 95% confidence interval. Bayesian multivariate generalized linear mixed model (MGLMM) was used to estimate the p-value. The Bayesian MGLMM was fitted with sociodemographic factors, BMI, Bacillus Calmette-Guérin (BCG) vaccination and preventive measures practiced as fixed effects and cluster effects were taken as a random effect.

nation-wide serosurvey in India. A recent publication from Bangladesh on seroprevalence of SARS-CoV-2 in a subdistrict of Chattogram also found a similar level of seroprevalence (64%) [37] comparing urban and rural populations.

Densely populated slums showed higher seropositivity (71.0%) than the neighboring non-slum areas (62.2%) [23]. Even though the numbers of family members living in the same household in slums (4.87) were found to be fewer than those in non-slum households (5.09), populaton density in slum is much higher because of the the congested living quarters (82% of the households possess a one-room dwelling space with a mean area of 119 sq ft). Other facilities are commonly shared in slums (e.g. water sources: 92%, latrine: 90% and cooking places: 60%) [30]. Positive association of seroprevalence with lower income and limited years of formal education may also explain the higher SARS-CoV-2 seropositivity in slum than non-slum populations.

Studies among hospitalized patients or patients recovering from COVID-19 have observed that older age, male gender, overweight/obesity and underlying chronic diseases are all associated with worse disease outcomes [27,28]. Our study, based in urban community settings showed that a great proportion of the seropositive population have been asymptomatic (62.4%) with fewer having mild COVID-like symptoms. Nonetheless, the same risk factors that were identified elsewhere in moderately or severely ill patients were found to be linked with higher odds of SARS-CoV-2 seropositivity. We found that irrespective of the living conditions, seropositivity was positively associated with, overweight,self-reported diabetes, heart disease or a history of stroke.

Practice of preventive measures, such as regular wearing of masks and washing hands were associated with lower seropositivity Our finding is in accord with a recently conducted randomized controlled trial from Bangladesh which demonstrated that promotion of increased use of masks reduced symptomatic SARS-CoV-2 infections in the community [38].

We found that BCG vaccinated people had lower probability of getting infected with SARS-CoV-2. There is no solid evidence yet that the BCG vaccination protects against the infection. A number of studies have described broad cross-protective effects of the BCG vaccine toward diverse unrelated infections, which is believed to be mediated through induction of trained immunity or long-term maintenance of innate immune memory [39,40]. The WHO does not endorse BCG vaccination to protect against COVID-19; however, it plans to assess the findings of the ongoing clinical trials addressing this query.

A number of studies have indicated possible relationship of physical activity with SARS-CoV-2 seropositivity. A study carried out in 48,440 adult patients analyzing the association between severe COVID-19 outcomes and self-reported physical activity demonstrated that patients consistently meeting physical activity guidelines were at lower risk of severe disease outcomes [41]. An inpatient- and outpatient-based study showed that individuals with lower levels of routine physical activity were affected by more severe forms of COVID-19 [42]. Among athletes, the maximal exercise capacity prior to infection was inversely associated with the likelihood of hospitalization due to COVID-19 [43]. Another study reported that physical activity at the county level was negatively associated with both COVID-19 cases and deaths per 100,000 residents in USA [44]. However, we did not find any association of mild-to-intense physical activity with SARS-CoV-2 seropositivity.

Our study has a number of limitations. The study is based on a community sample where the seropositive cases were mostly asymptomatic or mildly symptomatic. Thus, there was no scope to study association of disease severity with various factors (biological/behavioral/social). Data and samples were collected purposively from two large cities in Bangladesh focusing on slums and adjacent non-slum areas. This sampling was not based on primary sampling unit, which would have been representative of the cities. The participant enrolment from slums and

non-slums were not equal due to non-response, particularly from non-slum households during the COVID-19 pandemic. To overcome this bias due to non-response, a population-based weighted score was created and applied to estimate weighted seroprevalence of SARS-CoV-2 and also used as cluster effect in the logistic regression model [45]. The serosurvey was carried out over a long period (~5 months) during which the transmission levels could change, making it difficult to estimate the prevalence and understand its importance in relation to caseloads. However, during the period of December 2020 to February 2021, the rate of active infection (RT-PCR) was low [46], COVID-19 vaccination had not yet started and thus may have minimally affected the survey. Another limitation was that a shortened version of the physical activity questionnaire was applied that did not capture the full spectrum of activities. Establishing BCG vaccination status, in absence of vaccination cards required a visible vaccination scar on the upper arm, which may fade or disappear with time and thus vaccination status may be underrepresented. Self-reported COVID-like symptoms and preventive behaviors may suffer from recall bias.

In conclusion, the estimated SARS-CoV-2 antibody seroprevalence was higher in slum than in non-slum areas of two large cities of Bangladesh. Already identified risk factors for disease severity in clinical patients such as overweight, diabetes, heart disease, and stroke were also found to be associated with infection in the urban communities. The behavior of wearing masks and washing hands regularly seemed to have beneficial effect against SARS-CoV-2 infection. Future serosurveillance studies in the SARS-CoV-2 vaccine era should be planned to monitor exposure and side-by-side determine and differentiate between infection- and vaccine-induced humoral immunity. Such data will be crucial to inform public health decision-makers to improve vaccine distribution and allocation, and appraise booster dose requirements during the COVID-19 pandemic.

## Supporting information

**S1 Table. Internal validation of the Elecsys Anti-SARS-CoV-2 Immunoassay Kit.**
(DOCX)

**S2 Table. Weighted seroprevalence of SARS-CoV-2 antibodies among the residents of slum and non-slum neighborhoods of the Dhaka and Chattogram districts.**
(DOCX)

**S3 Table. Weighted seroprevalence of SARS-CoV-2 antibodies among the participants with self-reported occurrence of COVID-like symptoms in the past six months.**
(DOCX)

**S4 Table. Weighted seroprevalence of SARS-CoV-2 antibodies among the participants with co-morbidities.**
(DOCX)

**S1 File. Field data collection and study team.**
(DOCX)

**S2 File. Sample size.**
(DOCX)

## Acknowledgments

We warmly thank all of our participants for their consistently gracious welcome and sincere help during the study. We gratefully acknowledge the contribution of Mr. AHM Gulam

Mustafa who contributed to the Data management. We are thankful to Bangladesh Health Watch, our advocacy partner for their continued support throughout the study period.

## Author Contributions

**Conceptualization:** Rubhana Raqib, Fahmida Tofail, Abdur Razzaque.

**Data curation:** Md. Ahsanul Haq.

**Formal analysis:** Md. Ahsanul Haq.

**Funding acquisition:** Rubhana Raqib, Abdur Razzaque.

**Investigation:** Rubhana Raqib, Abdur Razzaque.

**Methodology:** Rubhana Raqib, Evana Akhtar, Anjan Kumar Roy, Md. Biplob Hosen, Farjana Haque.

**Project administration:** Rubhana Raqib.

**Resources:** Rubhana Raqib.

**Software:** Md. Ahsanul Haq.

**Supervision:** Rubhana Raqib, Protim Sarker, Evana Akhtar, Md. Razib Chowdhury, Abdur Razzaque.

**Validation:** Rubhana Raqib, Evana Akhtar.

**Visualization:** Rubhana Raqib.

**Writing – original draft:** Rubhana Raqib, Protim Sarker, Evana Akhtar.

**Writing – review & editing:** Rubhana Raqib, Tarique Mohammad Nurul Huda, Daniel D. Reidpath, Dewan Md. Emdadul Hoque, Zahirul Islam, Shehlina Ahmed, Tahmeed Ahmed, Fahmida Tofail, Abdur Razzaque.

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
