## [Decision Letter · Decision Letter 0]

20 Jan 2022

PONE-D-21-35521Seroprevalence of SARS-CoV-2 infection and associated factors among slum and non-slum dwellers in Bangladesh during October 2020 to February 2021PLOS ONE

Dear Dr. Raqib,

Thank you for submitting your manuscript to PLOS ONE. After careful consideration, we feel that it has merit but does not fully meet PLOS ONE’s publication criteria as it currently stands. Therefore, we invite you to submit a revised version of the manuscript that addresses the points raised during the review process.

We look forward to receiving your revised manuscript.

Kind regards,

Basant Giri, Ph.D.

Academic Editor

PLOS ONE

Journal Requirements:

3. PLOS requires an ORCID iD for the corresponding author in Editorial Manager on papers submitted after December 6th, 2016. Please ensure that you have an ORCID iD and that it is validated in Editorial Manager. To do this, go to ‘Update my Information’ (in the upper left-hand corner of the main menu), and click on the Fetch/Validate link next to the ORCID field. This will take you to the ORCID site and allow you to create a new iD or authenticate a pre-existing iD in Editorial Manager. Please see the following video for instructions on linking an ORCID iD to your Editorial Manager account: https://www.youtube.com/watch?v=_xcclfuvtxQ.

“This work was funded by The Foreign, Commonwealth & Development Office (FCDO) through The United Nations Population Fund (UNFPA), and Global Affairs Canada . icddr,b is also grateful to the Governments of Bangladesh, Canada, Sweden and the UK for providing core/unrestricted support for its operations and research. We warmly thank all of our participants for their consistently gracious welcome and sincere help during the study. We gratefully acknowledge the contribution of Mr Gulam Mustafa who contributed to the Data management.”

 “No”

Reviewers' comments:

Reviewer's Responses to Questions

**Comments to the Author**

1. Is the manuscript technically sound, and do the data support the conclusions?

Reviewer #1: Partly

Reviewer #2: Yes

2. Has the statistical analysis been performed appropriately and rigorously? 

Reviewer #1: No

Reviewer #2: Yes

3. Have the authors made all data underlying the findings in their manuscript fully available?

Reviewer #1: Yes

Reviewer #2: No

4. Is the manuscript presented in an intelligible fashion and written in standard English?

Reviewer #1: Yes

Reviewer #2: Yes

5. Review Comments to the Author

Reviewer #1: Rephrase line 91-92: The claims made in the sentence, that there have been limited surveys in LMICs aren't supported by the citations. In fact, the citations negate the point you are trying to make. Please rephrase the sentence or provide citations that corroborate your claim.

Line 102-103: Around 12.2 million people live in impoverished, densely populated urban centers of Dhaka and Chattogram areas (19-21)

Do the authors mean that the total population of Dhaka and Chittagong is only 12.2 million? If not please rephrase the sentence to say that 12.2 million people are impoverished in Dhaka and Chattogram (or something to that effect).

Line 104-105: "However, there are no data about the COVID situation in these communities."

It is unlikely there are not no data at all. There must have been RT-PCR tests done, even if inadequately. The authors probably mean there are limited data about the COVID-19 situation in these communities. And COVID-19 instead of COVID.

"SARS-CoV-2 serosurveys would estimate the prevalence of SARS CoV-2 infection in the communities."

A serosurvey will not estimate the prevalence of SARS-COV-2 infection. It will estimate the prevalence of individuals with a history of SARS-COV2 infection. Those two are not the same.

Line 107: "... associated with disease severity" do you mean COVID-19 severity? It is not clear from the sentence.

Line 108-109: "It would be important to find out whether different demographic characteristics and co-morbidities prevailing in the slums and non-slum urban areas are also associated with seroprevalence."

Line 104 and 107: Consider replacing "would be" with is.

Line 109: "... are also associated with seroprevalence."

What does this mean? Do you mean "are also associated with SARS-CoV-2 seropositivity? Same issue in some other instances (line 304, 306 for example).

Line 112: "... taking into account both the symptomatic and asymptomatic cases".

Please consider deleting this part. It will make the sentence more clear.

Line 214-215: "A weighted analysis was performed using the number of clusters from the eight selected areas and selected participants from the clusters to better reflect the current seroprevalence of SARS-CoV-2."

What does this mean? What was the weighting based on? Age? Sex? Other variables? It is not clear from the text. Please clarify explicitly.

Line 344-355: It is plausible that the lower probability of SARS-CoV-2 infection observed among the Bangladeshi participants, who carry out moderate physical activity may be linked to stronger immune status.

How is this plausible? Multiple jumps in logic. And there really is no such thing as a strong immune status; there is immunocompromised status, normal immune response and a hyper-immune response (which is a problem in its own, even in COVID-19).

Line 359-360:

".. physicians advise COVID patients to refrain from exercising when overt symptoms develop, and gradually return to "

Do they? Isn't this a casual blanket statement?

Other issues:

Was the test inaccuracy corrected for? The authors have focused on developing a regression model to evaluate predictors of seropositivity using a mixed effects model, however there was no effort in correcting test errors. Based on the authors' own validation studies, the sensitivity was around 93% and the specificity was between than 97.9%-100%. When these inaccuracies are corrected for, the final adjusted seroprevalence will be different from the one that has been presented in this study. Without them seroprevalence studies aren't really considered methodologically sound. Probabilistic models make it easy to correct these errors. Probabilistic models also make the weighting process easy. But such corrections can also be done without using a probabilistic model.

Please see here: https://rss.onlinelibrary.wiley.com/doi/full/10.1111/rssc.12435

here: https://ete-online.biomedcentral.com/articles/10.1186/1742-7622-9-9.

And here: https://www.thelancet.com/journals/lancet/article/PIIS0140-6736(20)31304-0/fulltext

If the authors are not going to correct for test inaccuracy, there should at least be a discussion about why they chose not to do so.

Some of the conclusions drawn from the study findings have problems with face validity (physical activity and seropositivity for example). There also appear to be obvious issues with confounding and identification. Therefore it would be prudent to tread these findings with caution and not read too much into them. Many of the predictors that the authors have tried to study in this observational study have been studied in experimental trials (The Bangladesh mask study for example https://www.science.org/doi/10.1126/science.abi9069) and it might be useful to discuss those findings as well.

Reviewer #2: The authors used standard English with a very clear flow, they also presented clear tables, graphs and additional information for reference. I did not encounter any errors while reviewing the manuscript.

6. PLOS authors have the option to publish the peer review history of their article (what does this mean?). If published, this will include your full peer review and any attached files.

Reviewer #1: No

Reviewer #2: No

---

## [Author Response · Author response to Decision Letter 0]

3 Apr 2022

Reviewers' comments:

Reviewer's Responses to Questions

Comments to the Author

1. Is the manuscript technically sound, and do the data support the conclusions?

Reviewer #1: Partly

Reviewer #2: Yes

2. Has the statistical analysis been performed appropriately and rigorously? 

Reviewer #1: No

Reviewer #2: Yes

3. Have the authors made all data underlying the findings in their manuscript fully available?

Reviewer #1: Yes

Reviewer #2: No

4. Is the manuscript presented in an intelligible fashion and written in standard English?

Reviewer #1: Yes

Reviewer #2: Yes

5. Review Comments to the Author

Reviewer #1: Rephrase line 91-92: The claims made in the sentence, that there have been limited surveys in LMICs aren't supported by the citations. In fact, the citations negate the point you are trying to make. Please rephrase the sentence or provide citations that corroborate your claim.

Response: As advised we have rephrased the sentence. The revised phrase now reads as-

“Population-based serosurveys for SARS-CoV-2 conducted in LMICs have been either national surveys (4, 5) or predominantly focused on frontline healthcare workers or industry workers (6-10) but few have studied marginalized populations in slums (11, 12).” (Lines 74-77).

Line 102-103: Around 12.2 million people live in impoverished, densely populated urban centers of Dhaka and Chattogram areas (19-21).

Do the authors mean that the total population of Dhaka and Chittagong is only 12.2 million? If not please rephrase the sentence to say that 12.2 million people are impoverished in Dhaka and Chattogram (or something to that effect).

Response: The revised sentence is as follows:

“Around 12.2 million impoverished people of Dhaka and Chattogram cities live in densely populated low-income urban slums.” Line: 86-87

Line 104-105: "However, there are no data about the COVID situation in these communities."

It is unlikely there are not no data at all. There must have been RT-PCR tests done, even if inadequately. The authors probably mean there are limited data about the COVID-19 situation in these communities. And COVID-19 instead of COVID.

Response: The time when the survey was carried out in Bangladesh, information about the COVID-19 situation in slums was largely missing, because majority of the slum dwellers are poor and they did not go to the hospital or clinic facilities for PCR testing because of high test price. Moreover, very long queues in public facilities and fear of wage loss/loss of income due to lockdowns imposed by city corporations over the slum areas or other neighborhoods made people reluctant to go for testing. We have revised the sentences as follows:

“So far limited data about the COVID situation in these slum communities are available from Bangladesh (22). Studies targeting RT-PCR-based detection of SARS-CoV-2 infection in Bangladesh have been mainly carried out in hospital settings (23-25); such data from the community is lacking.” Line: 87-90

"SARS-CoV-2 serosurveys would estimate the prevalence of SARS CoV-2 infection in the communities."

A serosurvey will not estimate the prevalence of SARS-COV-2 infection. It will estimate the prevalence of individuals with a history of SARS-COV2 infection. Those two are not the same.

Response: As correctly indicated by the Reviewer, a SARS-CoV-2 serosurvey will estimate the prevalence of cases/individuals with a history of SARS-CoV-2 infection in the communities. We have revised the sentence accordingly (Lines 90-93).

Line 107: "... associated with disease severity" do you mean COVID-19 severity? It is not clear from the sentence.

Response: We meant ‘severity of COVID-19’ (not severity of NCDs). The sentence has been revised accordingly (Line 95).

Line 108-109: "It would be important to find out whether different demographic characteristics and co-morbidities prevailing in the slums and non-slum urban areas are also associated with seroprevalence."

Line 104 and 107: Consider replacing "would be" with ‘is’.

Response: Corrected as advised (Line 95). 

Line 109: "... are also associated with seroprevalence."

What does this mean? Do you mean "are also associated with SARS-CoV-2 seropositivity? Same issue in some other instances (line 304, 306 for example).

Response: Yes, as indicated by the reviewer the correct phrase is “SARS-CoV-2 seropositivity or Seroprevalence of SARS-CoV-2”. We have made the necessary changes in several places (Line 133, 211, 245, 252, 262, 274, 277, 285, 290, 293, 295, 307, 309, 313, 321, 354). 

Line 112: "... taking into account both the symptomatic and asymptomatic cases".

Please consider deleting this part. It will make the sentence more clear.

Response: Done as suggested and sentence rephrased (Lines 101).

Line 214-215: "A weighted analysis was performed using the number of clusters from the eight selected areas and selected participants from the clusters to better reflect the current seroprevalence of SARS-CoV-2."

What does this mean? What was the weighting based on? Age? Sex? Other variables? It is not clear from the text. Please clarify explicitly.

Response: We have calculated population-based weighted prevalence on the basis of sum of two probabilities, between cluster probability (p1) and within cluster probability (p2). The sample was collected from 4 areas of Dhaka (Korail, Mirpur, Dhalpur and Ershad Nagar) and 2 areas of Chottagram (Shaheed Lane and Akbar Shah Kata Pahar,) that harbored both slum and surrounding non-slum areas. For each selected area, a total probability score was calculated as (p1+p2). The weight was determined as the inverse of total probability of each selected area (1/(p1+p2)). Thereafter, the calculated weight was distributed to all selected participants and the weighted prevalence was estimated. This description has been added in Statistical analysis section (Line 198-205).

The sum of probability scores of all selected areas was 1.00.

Line 344-355: It is plausible that the lower probability of SARS-CoV-2 infection observed among the Bangladeshi participants, who carry out moderate physical activity may be linked to stronger immune status.

How is this plausible? Multiple jumps in logic. And there really is no such thing as a strong immune status; there is immunocompromised status, normal immune response and a hyper-immune response (which is a problem in its own, even in COVID-19).

Response: According to published literature, “physical inactivity is associated with a higher risk for severe COVID-19 outcomes” or in other words “...meeting physical activity guidelines was strongly associated with a reduced risk for severe COVID-19 outcomes among infected adults” (Sallis R 2021). We found a positive association between seropositivity and moderate physical activity, but not with intense activity which was surprising.

However, based on the Reviewer’s suggestion below to correct for “test inaccuracy” a Bayesian multivariate generalized linear mixed model (MGLMM) was applied. This led to some changes in the results. The association of seropositivity with grades of physical activity no longer remained significant. Accordingly, revisions were made in the Results (Lines 301-302), Discussion (Lines 354-355 & 363-364). The figure on effect of physical activity on seroprevalence (previous Figure 3) has now been removed.

- Sallis R, Young DR, Tartof SY, Sallis JF, Sall J, Li Q, et al. Physical inactivity is associated with a higher risk for severe COVID-19 outcomes: a study in 48 440 adult patients. Br J Sports Med. 2021;55(19):1099-105.

Line 359-360:

".. physicians advise COVID patients to refrain from exercising when overt symptoms develop, and gradually return to "

Do they? Isn't this a casual blanket statement?

Response: We agree with the reviewer; this sentence has been deleted. 

Other issues:

Was the test inaccuracy corrected for? The authors have focused on developing a regression model to evaluate predictors of seropositivity using a mixed effects model, however there was no effort in correcting test errors. Based on the authors' own validation studies, the sensitivity was around 93% and the specificity was between than 97.9%-100%. When these inaccuracies are corrected for, the final adjusted seroprevalence will be different from the one that has been presented in this study. Without them seroprevalence studies aren't really considered methodologically sound. Probabilistic models make it easy to correct these errors. Probabilistic models also make the weighting process easy. But such corrections can also be done without using a probabilistic model.

Please see here: https://rss.onlinelibrary.wiley.com/doi/full/10.1111/rssc.12435

here: https://ete-online.biomedcentral.com/articles/10.1186/1742-7622-9-9.

And here: https://www.thelancet.com/journals/lancet/article/PIIS0140-6736(20)31304-0/fulltext

If the authors are not going to correct for test inaccuracy, there should at least be a discussion about why they chose not to do so.

Response: We very much appreciate the useful advice by the Reviewer. Accordingly, we have carried out relevant analysis in the manuscript and made necessary changes in the Tables, figures and texts. A paragraph has been added in the statistical description about the application of Bayesian theorem to fix the inaccuracy of the antibody test (Lines 212-216)

“Since the in-house validation of the antibody assay showed a sensitivity of about 93% and a specificity between 97.9%-100% (manufacturer reported sensitivity is 99% and specificity is 100%), to correct the test inaccuracy we estimated seroprevalence of SARS-CoV-2 associated risks (odds ratio) by Bayesian multivariate generalized linear mixed model (MGLMM).”

Some of the conclusions drawn from the study findings have problems with face validity (physical activity and seropositivity for example). There also appear to be obvious issues with confounding and identification. Therefore, it would be prudent to tread these findings with caution and not read too much into them. Many of the predictors that the authors have tried to study in this observational study have been studied in experimental trials (The Bangladesh mask study for example https://www.science.org/doi/10.1126/science.abi9069) and it might be useful to discuss those findings as well.

Response: We thank the reviewer for a useful and constructive comment. Based on the suggestion above to correct analysis for test inaccuracy, we applied Bayesian MGLMM. The revised analysis showed that individuals routinely wearing face masks and washing hands had lower odds of getting seropositive (Lines 296-300). This was in agreement with the findings of the Bangladesh Mask study and we referred to it (Abaluck J 2022). Accordingly, we have revised the findings in the Discussion (Lines 343-346). The Conclusion has also been revised (Lines 387-388). 

Furthermore, as advised, we have substantially reduced the sections on BCG vaccination and physical activity.

- Abaluck J, Kwong LH, Styczynski A, Haque A, Kabir MA, Bates-Jefferys E, et al. Impact of community masking on COVID-19: A cluster-randomized trial in Bangladesh. Science. 2022;375(6577):eabi9069.

Reviewer #2: The authors used standard English with a very clear flow, they also presented clear tables, graphs and additional information for reference. I did not encounter any errors while reviewing the manuscript.

6. PLOS authors have the option to publish the peer review history of their article (what does this mean?). If published, this will include your full peer review and any attached files.

Do you want your identity to be public for this peer review? For information about this choice, including consent withdrawal, please see our Privacy Policy.

Reviewer #1: No

Reviewer #2: No

Reviewer: Amos Hamukale

Manuscript:

Line 141-143: highlights that during lockdown accessibility to non-slum households was limited hence collected data from 2118 slum inhabitants and 1102 from non-slum in a 2:1 ratio. It is not clear how sample size was determined, it seems the sample size was adjusted to suit the non-response in households that could not be reached hence introducing systematic bias. 

Response: We thank the reviewer for raising this issue. The study was conducted in the early stage of COVID-19 in Bangladesh. During that time no published data was found about the prevalence of COVID-19. Thus, we conducted a pilot study in icddr,b’s HDSS area and calculated the sample size based on this finding. The non-slum participants frequently did not allow study teams to enter their household and the rejection rate was high. Thus, we draw the samples in a 2:1 ratio. To reduce the systematic bias, a population-based weighted score was created, which was applied to estimate weighted seroprevalence as well as use as cluster effects in the logistic regression model (Gelman A). Please also see response to Reviewer#1’s query about weighted analysis.

- Gelman A, Carpenter B. Bayesian analysis of tests with unknown specificity and sensitivity. Journal of Royal statistical Society Applied Statistics 2020;69(5):1269-83.

Line 210-211: The mean with standard deviation are reported for continuous data but there is not indication that the data was checked for normality which would determine whether to use mean or median.

Response: Main exposure of the study was dichotomous and the statistical model was performed with mixed effect logistic regression model thus there is no need to check the normality. 

Line 214-216 & 238: Authors indicated that they conducted a weighted analysis but did not provide adequate information on what kind of weights were used for example Design weights or post-Stratification weights for the different variables.

Response: We have calculated population-based weighted prevalence on the basis of sum of two probabilities, between cluster probability (p1) and within cluster probability (p2). The sample was collected from 4x2 areas of Dhaka (Korail, Mirpur, Dhalpur and Ershad Nagar) and 2x2 areas of Chottagram (Shaheed Lane and Akbar Shah Kata Pahar,) that harbored both slum and corresponding non-slum areas. For each selected area, a total probability score was calculated as (p1+p2). The weight was determined as the inverse of total probability of each selected area (1/(p1+p2)). Thereafter, the calculated weight was distributed to all selected participants and the weighted prevalence was estimated. This description has been added in Statistical analysis section (Line 198-205).

Table 1: Age in years should be categorized to better understand the distribution and how they affect the analysis. Having more children in one group can easily skew the data hence giving a false average.

Response: As suggested, we have included the number of participants in each age category in Table 1 showing the distribution in slum and non-slum areas. Additionally, for the reviewer, we have provided a table below showing age-wise distribution in seropositive and seronegative categories.

Age category, years Overall Number Seropositive Seronegative

10-17 years 776 508 268

18-30 years 945 649 296

31-50 years 1015 715 300

>50 years 484 337 147

Total 3220 2209 1011

Table 1: Authors have indicated p-value in line 248-249 but these are not reported in the table.

Response: We thank the reviewer for indicating the mistake. The sentence has been deleted from the footnote of Table 1.

Table 3: On years of education, the authors use 11-15 years as the reference group. This reference group does not adequately represent the Slum dwellers 3% hence reduces the precision.

Response: The Reviewer has correctly indicated the drawback. For slum dwellers, the reference group will be 6-10 years of education. This has been changed in the Table 3.

---

## [Editor Report · Decision Letter 1]

20 Apr 2022

PONE-D-21-35521R1Seroprevalence of SARS-CoV-2 infection and associated factors among Bangladeshi slum and non-slum dwellers in pre-COVID-19 vaccination era: October 2020 to February 2021PLOS ONE

Dear Dr. Raqib,

The editorial manager system can now allow you to upload your updated revised manuscript. Therefore, we invite you to submit a revised version of the manuscript that addresses the points raised during the review process.

We look forward to receiving your revised manuscript.

Kind regards,

Basant Giri, Ph.D.

Academic Editor

PLOS ONE
---

## [Author Response · Author response to Decision Letter 1]

21 Apr 2022

Comment on 29 March 2022

-We note you updated your Data Availability statement to the following:

- "Yes - all data are fully available without restriction"

- "All datasets used in this study are available."

While you have stated that the data is fully available without legal or ethical restrictions, you have not explained where the data can be found and how others may access the data.

Response: We have added the Data Availability statement describing the compliance with PLOS' data policy in the MS (in data availability section line number:405-409) as well as in the MS submission system. The revised manuscript was submitted on 3rd April 2022.

Comment on 13 April 2022

-We are requesting for an updated version of your manuscript including the ethical approval number provided by the institutional review board.

Response: As requested, we have added the ethical approval number (PR-20070, dated 1st September 2020) provided by the Ethical Review Committee (ERC) of icddr,b in the manuscript Lines 193-196. However, it was not possible to upload the revised manuscript in the submission system, as the system did not allow uploading. Therefore, we replied to the mail by attaching the revised manuscript. 

Comment on 20th April 2022

-The editorial manager system can now allow you to upload your updated revised manuscript. Therefore, we invite you to submit a revised version of the manuscript that addresses the points raised during the review process. Please include the following items when submitting your revised manuscript:

• An unmarked version of your revised paper without tracked changes. You should upload this as a separate file labeled 'Manuscript'

Response: There are no further comments from the reviewers after submission of the revised manuscript and response to Reviewers’ comments on 10th February 2022. The rebuttal letter that responds to point raised by the academic editor and Managerial Desk have been addressed in the file labeled as 'Response to Reviewers'.

As indicated by the Academic Editor, we have now uploaded a revised copy of the manuscript with revision marked in track changes, labeled as “Revised Manuscript with Track Changes” and a clean unmarked copy of the revised manuscript labelled as “Manuscript”.

---

## [Editor Report · Decision Letter 2]

22 Apr 2022

Seroprevalence of SARS-CoV-2 infection and associated factors among Bangladeshi slum and non-slum dwellers in pre-COVID-19 vaccination era: October 2020 to February 2021

PONE-D-21-35521R2

Dear Dr. Raqib,

We’re pleased to inform you that your manuscript has been judged scientifically suitable for publication and will be formally accepted for publication once it meets all outstanding technical requirements.

Kind regards,

Basant Giri, Ph.D.

Academic Editor

PLOS ONE

---

## [Editor Report · Acceptance letter]

28 Apr 2022

PONE-D-21-35521R2 

Seroprevalence of SARS-CoV-2 infection and associated factors among Bangladeshi slum and non-slum dwellers in pre-COVID-19 vaccination era: October 2020 to February 2021 

Dear Dr. Raqib:

I'm pleased to inform you that your manuscript has been deemed suitable for publication in PLOS ONE. Congratulations! Your manuscript is now with our production department. 

Kind regards, 

on behalf of

Dr. Basant Giri 

Academic Editor

PLOS ONE